# Creation of a non-Western humanized gnotobiotic mouse model through the transplantation of rural African fecal microbiota

Kristin M. Van Den Ham,[1] Morgan R. Little,[1] Olivia J. Bednarski,[1] Elizabeth M. Fusco,[1] Rabindra K. Mandal,[1] Riten Mitra,[2] Shanping Li,[3] Safiatou Doumbo,[4] Didier Doumtabe,[4] Kassoum Kayentao,[4] Aissata Ongoiba,[4] Boubacar Traore,[4] Peter D. Crompton,[3] Nathan W. Schmidt[1]

**ABSTRACT** Gut microbiota are increasingly being recognized as a contributing factor in the etiology of numerous diseases and as a potential determinant in the immune response to various treatments. Recent work has suggested that the suboptimal immunogenic response to vaccination in low- and middle-income countries may be associated with differences in the gut microbiome, which are known to be substantially different between Western and non-Western countries. However, insufficient consideration has been given to the characterization of non-Western microbiomes and their relationship with well-being and immunity. Humanized gnotobiotic mouse models have been used to better understand the causal associations between the gut microbiota and health outcomes but have largely been limited to the study of Western microbiota. Thus, we were interested in determining the applicability of gavage strategies used to humanize germ-free mice with Western microbiota to the humanization of germ-free mice with rural African fecal samples. Here, we assessed the impact of the number and frequency of gavages and the effect of a donor-matched diet on the colonization of Malian fecal microbiota in germ-free mice. One gavage was insufficient to provide a stable establishment of the Malian microbiome, whereas four weekly gavages resulted in a more consistent colonization of the human donor taxa. Interestingly, the donor-matched diet did not improve colonization over the fixed-formula, grain-based mouse chow. Subsequent phenotypic studies using African gut microbiota-humanized gnotobiotic mouse models will allow for a better understanding of the interaction between African gut microbiota and well-being and potentially aid in developing improved treatments for microbiota-dependent diseases in non-Western populations.

**IMPORTANCE** There is increasing evidence that microbes residing within the intestines (gut microbiota) play important roles in the well-being of humans. Yet, there are considerable challenges in determining the specific role of gut microbiota in human diseases owing to the complexity of diverse internal and environmental factors that can contribute to diseases. Mice devoid of all microorganisms (germ-free mice) can be colonized with human stool samples to examine the specific contribution of the gut microbiota to a disease. These approaches have been primarily focused on stool samples obtained from individuals in Western countries. Thus, there is limited understanding as to whether the same methods used to colonize germ-free mice with stool from Western individuals would apply to the colonization of germ-free mice with stool from non-Western individuals. Here, we report the results from colonizing germ-free mice with stool samples of Malian children.

**KEYWORDS** germ free mice, intestinal colonization, non-Western stool

Address correspondence to Nathan W. Schmidt, nwschmid@iu.edu.

The authors declare no conflict of interest.

See the funding table on p. 14.

Gut microbiota are increasingly being recognized as a contributing factor in the etiology of numerous diseases, including type 2 diabetes (1), inflammatory bowel disease (2, 3), multiple sclerosis (4, 5), and rheumatoid arthritis (6, 7), as well as diseases caused by intestinal and extraintestinal pathogens, including *Clostridium difficile* (8), influenza (9, 10), and *Plasmodium* (11, 12). Additionally, recent work has suggested that the microbiome composition may play a role in vaccine immunogenicity (13–15) and that differences in the gut microbiome may contribute to the substandard efficacy of vaccination in non-Western countries (16). Directly addressing the causality of gut microbiota toward the immune response in humans is challenging due to various confounding factors. Gut microbiota-humanized mice thereby provide an approach to establish potential causality by allowing control over the host genotype, diet, and housing conditions. However, the studies conducted so far to optimize the engraftment of human microbiota in germ-free or microbiota-depleted mice have only used Western fecal samples (17–21).

Gut microbiome humanization of conventional mice following microbiota depletion by either broad-spectrum antibiotics or bowel cleansing has demonstrated that a single gavage may be insufficient to establish successful and persistent engraftment of Western human gut microbiota in mice (17, 18). Moreover, certain keystone bacteria in the human gut microbiome, such as *Faecalibacterium prausnitzii*, may require multiple weekly gavages to be detected (18). However, it was also suggested that multiple gavages over too short a time period may perturb the stability of the newly forming gut microbiota (18). Interestingly, the gut microbiota composition of colonized germ-free and antibiotic-treated mice receiving one gavage (19) or five consecutive daily gavages (21) from the same Western human donor were found to be significantly different, which may have been the result of the different baseline intestinal environments in the microbiota-depleted conventional mice versus the germ-free mice (20), or possibly due to the bacterial communities remaining after antibiotic treatment hindering the engraftment of human gut microbiota.

Substantial differences in the composition of the gut microbiota are known to exist between industrialized and non-industrialized countries (15). Gut microbiota of rural African populations are lower in *Firmicutes* and enriched for *Bacteroidetes* compared with the microbiome of Western populations, and within the *Bacteroidetes* phylum itself, the African gut microbiome is predominately *Prevotella*, whereas the Western gut microbiome is predominately *Bacteroides* (22). Dietary differences are considered to be one of the main factors shaping the gut microbiome composition (22). Western diets are associated with a loss of *Prevotella* and fiber degradation capability (23) and an enrichment in *Verrucomicrobia* and mucin-utilizing glycoside hydrolases (24). Additionally, diet can significantly change gut microbiota composition of humanized mice (25, 26). Due to compositional differences, the humanization strategies characterized for Western gut microbiomes may not be transferable to non-Western microbiomes, and the use of a donor-matched diet may be crucial to achieve successful engraftment.

Herein, we examined the impact of the overall number and frequency of gavages on the engraftment efficacy of a rural Malian fecal microbiota in germ-free mice and the necessity of using a donor-matched diet to establish adequate colonization. Overall, we observed that the gavage strategies used for Western microbiomes were largely applicable to the Malian samples; a single gavage was insufficient to establish a stable gut microbiota, whereas multiple gavages resulted in colonization that more closely resembled that of the human donors. Moreover, we determined that using a donor-matched diet was unnecessary to ensure successful engraftment. Importantly, while we noted that rarer species were generally less successful at colonizing germ-free mice, certain taxa typically found at a low abundance in non-Western microbiota, particularly *Bacteroides*, greatly expanded within the mice, which may impact subsequent phenotypic studies. Improving the engraftment of non-industrialized microbiota in germ-free mice may aid in phenotype recapitulation, thereby allowing a better understanding of

the relationship between non-Western microbiota and health, and enhanced efficacy of treatments utilized in these populations.

## RESULTS

Comparison of humanization strategies in germ-free and microbiota-depleted mice using Western samples has generally demonstrated that one gavage is insufficient. Instead, multiple gavages, spaced out over multiple weeks to avoid perturbation of the newly forming microbiome, are likely the best strategy to successfully colonize human microbiota in mice (17, 18, 21). Moreover, diet can significantly influence the microbiome composition in humanized mice (25, 26). However, non-Western microbiomes have very different compositions; therefore, we were interested in confirming whether the same engraftment strategies would be applicable to a non-Western fecal sample in mice fed a donor-matched diet [Malian diet (MD)] (Fig. 1A). The human stool microbiome input consisted of five randomly selected fecal samples combined in equal amounts to better capture the potential diversity of the fecal microbiota in the Malian study population.

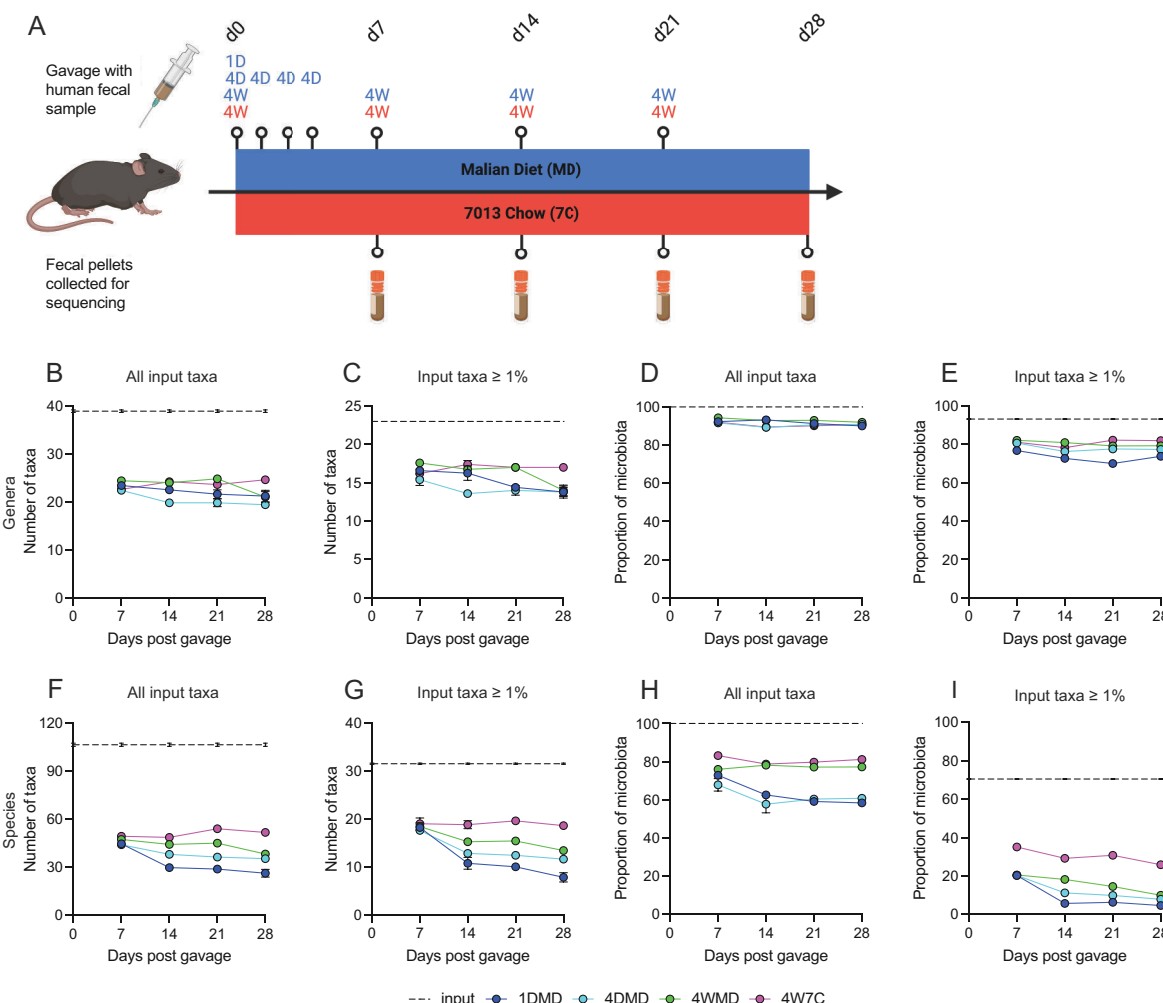

**FIG 1** Overview of humanization of germ-free mice. Regimen of human fecal microbiota engraftment (A). 1D = one gavage at the start of the experiment, 4D = four gavages on the first 4 consecutive days of the experiment, and 4W = four gavages at weekly intervals starting on the first day of the experiment. Blue indicates that the mice were fed the Malian diet and red indicates that the mice were fed Inotiv/Envigo 7013 mouse chow (7C). The number of genus-level taxa (B and C) and the proportion of microbiome accounted for by those taxa (D and E) in each mouse group over time. Genera found in at least one input sample (B and D) and genera present at an average of ≥1% in the input samples (C and E). The number of species-level taxa (F and G) and the proportion of microbiome accounted for by those taxa (H and I) in each mouse group over time. Species found in at least one input sample (F and H) and species present at an average of ≥1% in the input samples (G and I). Mean values ± SEM are plotted. Fig. 1A was created using BioRender.com.

One group of mice received one gavage at the start of the experiment (1DMD), a second group of mice received four gavages on the first 4 consecutive days of the experiment (4DMD), and a third group of mice received four gavages at weekly intervals starting on the first day of the experiment (4WMD). We were additionally interested in determining if it would be possible to successfully engraft the Malian fecal sample using a fixed-formula, grain-based diet, as opposed to the donor-matched diet, given that unlike the average Western diet, the typical diet at the study site in Mali is also low fat and polysaccharide based (Table S1A). Since previous studies using Western fecal samples suggested that the best engraftment strategy would be four gavages at weekly intervals, a fourth group of mice received four gavages at weekly intervals starting on the first day of the experiment (4W7C) but were provided with NIH-31 Modified Open Formula Mouse/Rat Sterilizable Diet (Inotiv/Envigo 7013; 7C), a fixed-formula, grain-based chow, rather than the donor-matched diet. Fecal pellets were collected weekly from each of the mice for 4 weeks, starting 1 week after the initial gavage.

The number of genus-level taxa that colonized the mice relative to the number detected in the input and the number found at ≥1% in the input suggests that rarer taxa generally had less successful engraftment within the mice (Fig. 1B and C). This trend was also observed with the species-level taxa (Fig. 1F and G). Additionally, there were more pronounced differences between the different engraftment groups at the species level. The total number of input species that colonized the mice was highest in 4W7C, followed by 4WMD and 4DMD, and was lowest in 1DMD. The majority of the mouse microbiome in all groups consisted of genus-level taxa detected in the input ($91.47 \pm 1.91\%$; Fig. 1D) and at ≥1% in the input ($78.19 \pm 4.14\%$; Fig. 1E). The percentage of the mouse microbiome consisting of all species present in the input was much higher than the proportion of the mouse microbiome consisting of species-level taxa found at ≥1% in the input (Fig. 1H and I), suggesting that there was an expansion of rarer input species within the mice. Furthermore, the differences between the engraftment groups were again more pronounced at the species level compared with the genus level. A larger percentage of the 4WMD ($77.27 \pm 2.60\%$ on d28) and 4W7C ($81.18 \pm 2.87\%$ on d28) microbiomes, compared with the 1DMD ($58.44 \pm 3.78\%$ on d28) and 4DMD ($60.85 \pm 3.95\%$ on d28) microbiomes, consisted of species detected in the input, and a larger percentage of the 4W7C microbiome consisted of species detected at ≥1% in the input compared with the Malian diet groups ($25.88 \pm 1.00\%$ compared with $9.93 \pm 1.19\%$ for 4WMD, $7.91 \pm 2.17\%$ for 4DMD, and $4.60 \pm 3.75\%$ for 1DMD on d28). Overall, the data suggest that while rarer species were generally less successful at colonizing the mice, some rarer species capable of engraftment were able to substantially expand within the mice to a greater extent than species that were initially more prevalent in the input.

Additionally, the data indicated that a proportion of the murine microbiome consisted of taxa that were not detected in the human input (Fig. S1). A greater proportion of the 1DMD and 4DMD microbiomes consisted of species not detected in the human input compared with the 4WMD and 4W7C microbiomes. The taxa identified in the mice, but not in the human input, are possibly the result of the expansion of specific taxa within the mice that were present in the human input but were below the detection level. The majority of the taxa detected in the Malian diet were either not detected in the mice or detected at low levels (< 1%). *Lactobacillus johnsonii* was the only species identified in the diet that was ≥1% in the mice, but *L. johnsonii* was only maintained at a level ≥1% following d7 in the 1DMD group (Tables S1B, S2 and S3).

Alpha diversity, as measured using observed features (richness/number of taxa present), Pielou evenness (evenness), Shannon diversity (richness and evenness), and Faith PD (phylogenetic diversity), was generally similar between the groups. However, the richness and evenness of the 1DMD group were consistently lower than those of the 4W7C group at later time points (Fig. S2). Using principal coordinate analysis (PCoA) based on Bray-Curtis distance, the microbiome of the mice fed the Malian diet grouped together on day 7 regardless of the gavage regimen (Fig. 2A through C). Following the initial time point, the 1DMD and 4DMD microbiomes became distinct from the 4WMD

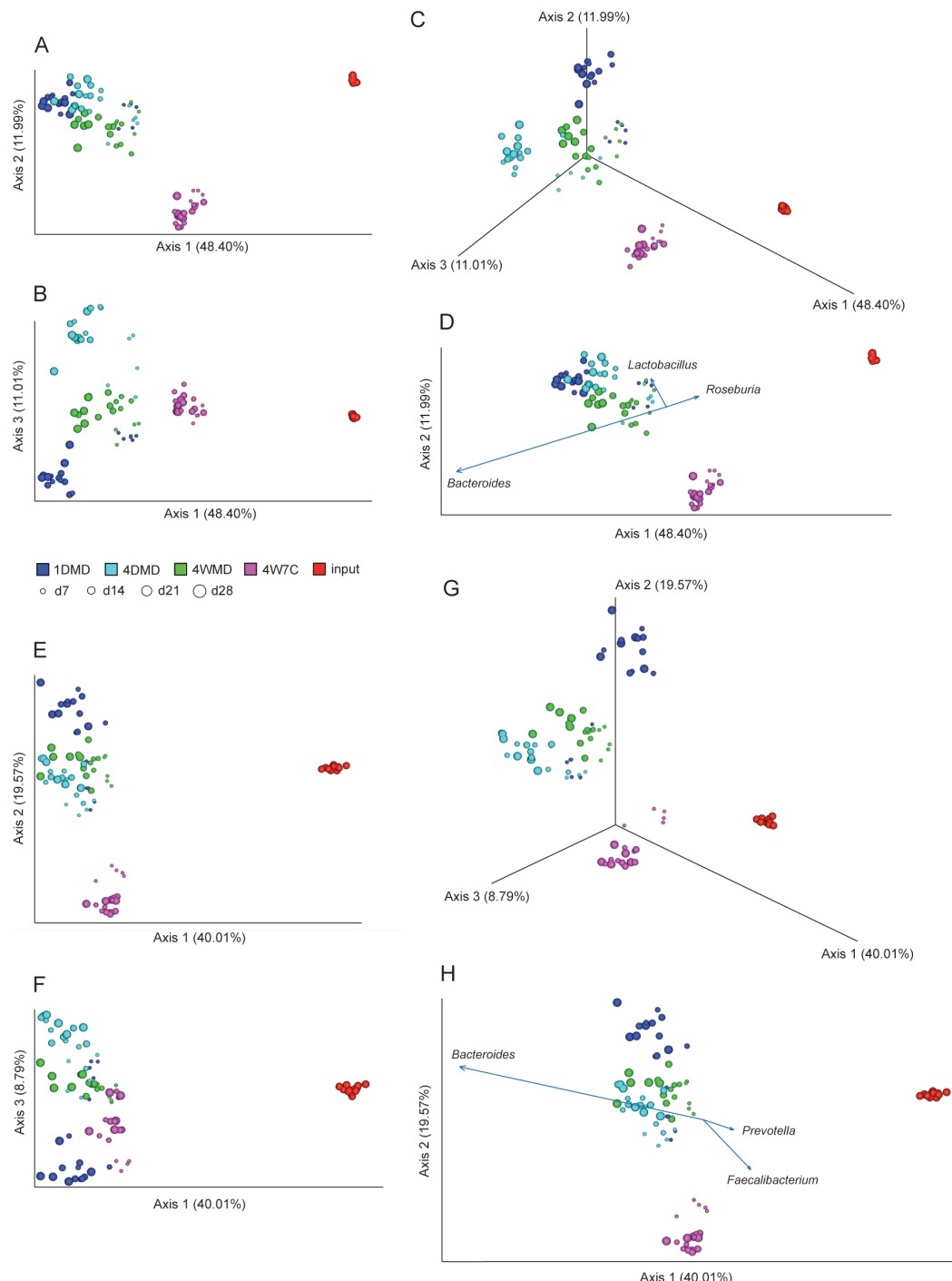

**FIG 2** Longitudinal beta-diversity of gut microbiome. PCoA plots based on Bray-Curtis distance (A–C) and unweighted Unifrac distance (E–G). Biplot of the top three genera driving ordination using Bray-Curtis distance (D) and unweighted Unifrac distance (H).

microbiome and from each other, but the microbiome composition at subsequent time points clustered together within each respective gavage group. The microbiome of the 4W7C mice was separate from the other groups but clustered together at all time points. Similar trends were observed with the Jaccard distance (Fig. S3A through C). When taking phylogeny into account using the unweighted (Fig. 2E through G) and the weighted (Fig. S3E through G) Unifrac distances, the 1DMD, 4DMD, and 4WMD microbiomes grouped together on the day 7 time point but subsequently diverged. The

4DMD and 4WMD microbiomes clustered together at the later time points, while the 1DMD microbiome clustered separately. The 4W7C microbiome was separate from the other groups at all time points. *Bacteroides* spp. were shown to be the dominant driver of ordination for all beta-diversity indices mentioned above (Fig. 2D and H; Fig. S3D and S3H), while *Lactobacillus* spp. and *Roseburia* spp. were also major drivers of ordination using Bray-Curtis, Jaccard, and weighted Unifrac distances (Fig. 2D; Fig. S3D and S3H), and *Faecalibacterium* spp. and *Prevotella* spp. were majors drivers of ordination using unweighted Unifrac distance (Fig. 2H).

 *Bacteroides* spp. were a major genus-level driver of ordination by all four beta-diversity metrics (Fig. 3A). *Bacteroides* spp. were detected at a low level in the input (Fig. 3A) but substantially expanded in all mouse groups; however, the composition and magnitude of *Bacteroides* spp. within the mice were dependent on the engraftment group. At the species level, *Bacteroides vulgatus*, *Bacteroides uniformis*, *Bacteroides nordii*, *Bacteroides ovatus*, and *Bacteroides intestinalis* (Fig. S4A, 4C through F) were also major drivers of ordination for all four beta-diversity metrics, while *Bacteroides thetaiotaomicron* (Fig. S4G) was a major driver for Bray-Curtis distance, unweighted Unifrac distance,

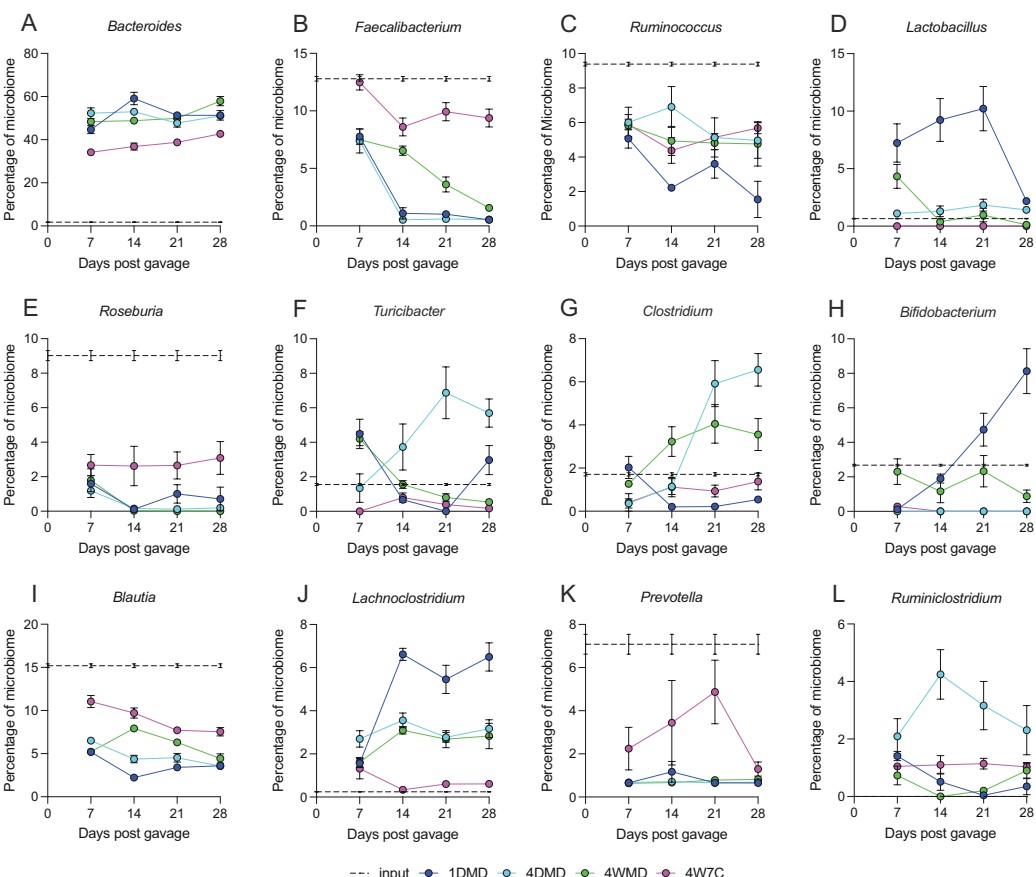

**FIG 3** Principal genera driving beta-diversity ordination. The top 10 genera driving ordination were determined using Jaccard distance, Bray-Curtis distance, unweighted Unifrac distance, and weighted Unifrac distance. *Bacteroides* spp., *Faecalibacterium* spp., *Ruminococcus* spp., *Lactobacillus* spp., and *Roseburia* spp. were major drivers for all four metrics (A–E). *Turicibacter* spp. and *Clostridium* spp. (F and G) were major drivers for Bray-Curtis distance, unweighted Unifrac distance, and weighted Unifrac distance. *Bifidobacterium* spp. was a major driver for Jaccard distance, Bray-Curtis distance, and unweighted Unifrac distance (H). *Blautia* spp. was a major driver for unweighted Unifrac distance and weighted Unifrac distance (I). *Lachnoclostridium* spp. was a major driver for Bray-Curtis distance and weighted Unifrac distance (J). *Prevotella* spp. was a major driver for Bray-Curtis distance and unweighted Unifrac distance (K). *Ruminiclostridium* spp. was a major driver for Jaccard distance and weighted Unifrac distance (L). *Dorea* spp., *Parabacteroides* spp., and *Eubacterium* spp. were only major drivers using Jaccard distance and are not shown. Mean values ± SEM are plotted.

and weighted Unifrac distance, and *Bacteroides caccae* (Fig. S4K) was a major driver for Bray-Curtis distance and unweighted Unifrac distance. *B. vulgatus* was highly prevalent in all groups, particularly in 1DMD (Fig. S4A), while *B. uniformis* was found at a higher percentage in the group fed the fixed-formula mouse chow (Fig. S4C), and *B. thetaiotaomicron* was found at a higher percentage in the groups fed the Malian diet (Fig. S4G). Interestingly, although *B. intestinalis* was the most prevalent *Bacteroides* species in 4DMD, it was barely detectable in the other groups (Fig. S4F). Furthermore, the percentage of the microbiome consisting of *B. nordii* and *B. caccae* continued to increase in 4WMD at later time points relative to the other groups (Fig. S4D and S4K). *F. prausnitzii* was a major driver of ordination by the four beta-diversity metrics and was detected in all groups (Fig. 3B; Fig. S4B) but substantially decreased over time in the mice fed the Malian diet. In contrast, the level of *F. prausnitzii* in the 4W7C group remained comparable to that of the input at all time points. *Ruminococcus* spp., *Roseburia* spp., *Blautia* spp., and *Prevotella* spp. (Fig. 3C, E, I and K) were all higher in the input compared with the murine groups, and in the case of *Roseburia* spp., *Blautia* spp., and *Prevotella* spp., their level in the 4W7C group was closer to the input than their level in the other groups. *L. johnsonii*, *Bifidobacterium pseudolongum*, and *Lachnoclostridium* spp. were substantially elevated in the 1DMD microbiome (Fig. 3D, H and J; Fig. S4H and S4I), while *Turicibacter* sp. LA61, *Clostridium* spp., and *Ruminiclostridium* spp. were all higher in the 4DMD microbiome relative to the other groups (Fig. 3F, G and L; Fig. S4J).

Previous studies examining ecological succession during the conventionalization of germ-free mice have shown that mature bacterial communities do not automatically reassemble in their original structure within recipient mice, but instead transition through various pioneering communities that make the intestinal environment more suitable for fastidious community members (27, 28). Choo and Rogers noted that fecal microbiota achieved compositional stability within 28 days of conventionalization (28), which is broadly in agreement with humanization studies (17, 18, 25). Consequently, we were interested in determining when the microbiome composition stabilized, as this would indicate the time required for engraftment before subsequent phenotypic studies should be conducted. According to all four beta-diversity indices, the microbiome generally stabilized between d21 and d28, although, according to some of the beta-diversity metrics, the microbiome of the 1DMD and 4WMD mice did not appear to stabilize in the time frame of the experiment (Fig. 4; Table S4). The overall distance between d7 and d28 was generally significantly larger for 1DMD compared with the other groups, and the distance was generally markedly smaller for the 4DMD and 4W7C groups (Fig. S5). Additionally, we investigated the intragroup distances at each time point to determine if the microbiome variability within each respective group changed over time. Intragroup variation measured by all four beta-diversity metrics remained largely the same for all groups throughout the experiment (Fig. S6; Table S5), likely due, at least in part, to coprophagy, as all mice within each respective group were housed together. The intragroup distance at the end of the experiment was the highest for group 1DMD, but this difference was only significant using Jaccard distance and unweighted Unifrac distance (Fig. S7).

Subsequently, the average distance to the input was examined (Fig. 5) to determine which regimen resulted in a microbiome engraftment closest to the starting material. The average distance to input increased significantly at later time points compared with d7 for the 1DMD and 4WMD groups as measured by all four beta-diversity matrices, and the average distance to input increased significantly for the 4DMD group as measured by the Jaccard distance and Bray-Curtis distance (Table S6). The average distance to the input did not significantly change for the 4W7C group throughout the experiment (Table S6). The average distance to the input at the d28 time point was consistently significantly lower for the 4W7C group compared with the other gavage groups (Fig. 6). These results demonstrate that the 4W7C gavage regimen resulted in a murine microbiome closest to the human input microbiome. In contrast, the average distance from the input to 1DMD was significantly higher than the distance to the other groups (by all four beta-diversity

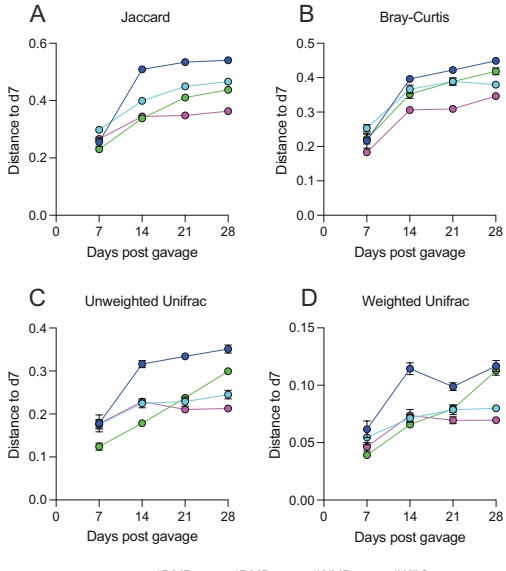

**FIG 4** Longitudinal stabilization of the microbiome. The Jaccard distance (A), Bray-Curtis distance (B), unweighted Unifrac distance (C), and the weighted Unifrac distance (D) from each time point compared with d7 within each group. Mean values ± SEM are plotted.

metrics for 4W7C, by Jaccard and Bray-Curtis distance for 4WMD, and by Jaccard distance for 4DMD), indicating that the murine microbiome resulting from a single gavage was the least similar to the starting human microbiome (Fig. 6A and B).

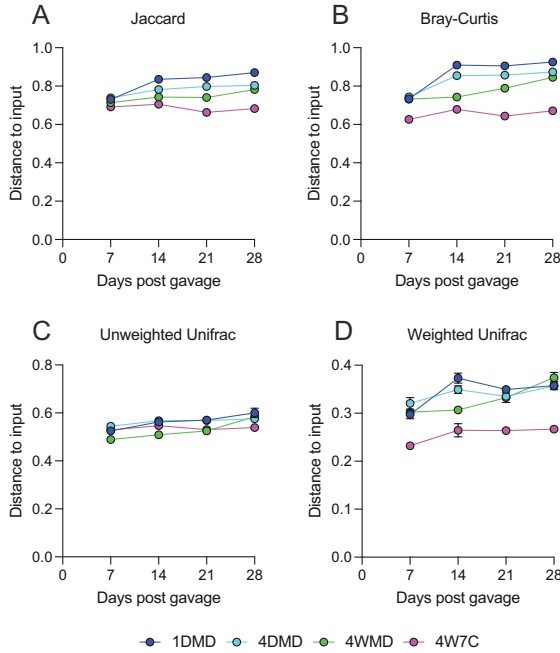

**FIG 5** Longitudinal distance to input. The average Jaccard distance (A), Bray-Curtis distance (B), unweighted Unifrac distance (C), and the weighted Unifrac distance (D) compared with the input within each group at each time point. Mean values ± SEM are plotted.

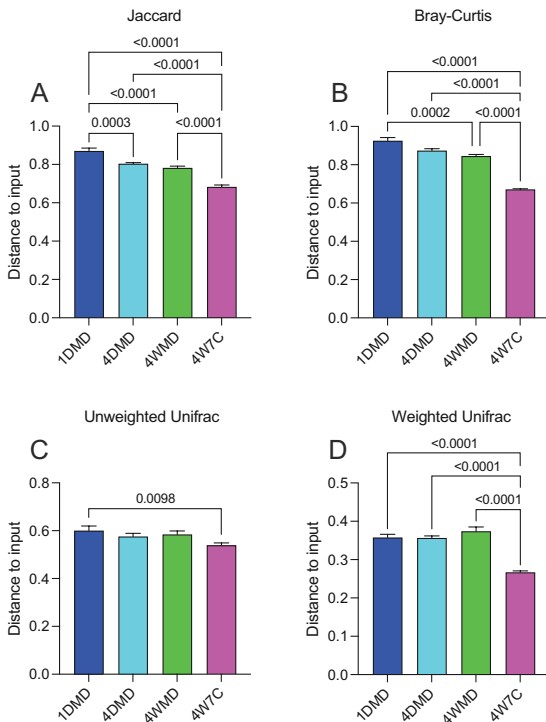

**FIG 6** Average distance to input on day 28. The average distance to the input for each group at d28 using Jaccard distance (A), Bray-Curtis distance (B), unweighted Unifrac distance (C), and the weighted Unifrac distance (D) was compared using ANOVA with Tukey's multiple comparison test. Mean values ± SEM are plotted.

## DISCUSSION

The predominant difference between the human input and the microbiota that engrafted in the mice was the expansion of *Bacteroides* (Fig. 2D and H; Fig. S3D and S3H). The relative proportion of the microbiome composed of the genus *Bacteroides* increased substantially in each mouse group (42.68%–57.90%) compared with that of the human input (1.76%) (Fig. 3A); however, the resulting composition and magnitude of *Bacteroides* within the mice were dependent on the engraftment group.

Many *Bacteroides* species encode a cytochrome *bd* oxidase, which has been hypothesized to be capable of reducing intestinal oxygen levels, thus allowing the obligate anaerobes to expand in the presence of oxygen (29, 30). Their capacity to tolerate and decrease oxygen levels and their proficiency in metabolizing many plant- and host-derived glycans (30, 31) may explain why they have been observed to expand in germ-free mice, not only by our lab but in many other humanization studies. Wos-Oxley *et al*. observed an expansion of *Bacteroides* in germ-free mice (61.2%) compared with the human donor (6.3%) (21); however, when antibiotic-treated conventional mice were gavaged with the same donor, a more muted increase in *Bacteroides* was observed (23.5%) (21). Staley *et al*. also found an increased frequency of *Bacteroides* in germ-free mice (25.37%) compared with antibiotic-treated conventional mice (13.75% and 11.39%, dependent on the antibiotic regimen used), but the abundance of *Bacteroides* spp. in the human donor was not noted (19). Furthermore, Lundberg *et al*. identified a substantial increase in *Bacteroides* spp. when germ-free mice (39%) were colonized with a human fecal sample (1%), but a similar expansion was not observed when germ-free mice were colonized with the microbiota of conventional mice (32). These differences may stem from the relative fitness of human and mouse *Bacteroides* in germ-free mice or from human *Bacteroides* being outcompeted by taxa persisting in the microbiota-depleted mice, such as *Muribaculaceae* (32), which are major utilizers of mucus-derived

monosaccharides (33). Interestingly, Turnbaugh *et al*. observed a similar increase in *Bacteroides* from the human donor (11.1%) to the recipient germ-free mice when fed a low-fat polysaccharide-rich mouse chow (22.5%), but the frequency of *Bacteroides* was subsequently reduced when the mice were switched to a Western diet (12.1%) (25).

The expansion of *Bacteroides* is particularly pertinent to the humanization germ-free mice with rural African fecal samples, as this genus is typically found at relatively low abundances in non-industrialized populations (22, 34–37). Thus, their expansion would increase the dissimilarity of the engrafted microbiome to a greater extent compared with that of a Western microbiome input. Consequently, it will need to be determined if the expansion of this genus prevents phenotype recapitulation in humanized mice or whether the presence of other key gut microbiota will be sufficient. If the expansion of *Bacteroides* inhibits phenotype manifestation, it may be possible to modulate the frequency of this taxa within the mice using diet (25). However, there seems to be a mismatch between diets and *Bacteroides* frequency; while Western diets seem to decrease *Bacteroides* prevalence in humanized mice (25), humans that eat a Western diet typically have a higher percentage of *Bacteroides* (34). Further investigation into which specific components of the Western diet are responsible for modulating *Bacteroides* frequency (38–41) may allow for the adjustment of this taxa without needing to use a diet diametric to that of the human donor.

Similarly, in this study, we were interested in determining whether the use of a fixed-formula, grain-based mouse chow (7013) would also allow the successful colonization of germ-free mice with the Malian microbiome, given that the donor-matched Malian diet (Table S1A) and the fixed-formula mouse chow are both low fat and polysaccharide rich. Previous work has indicated that diet is capable of quickly and significantly modulating gut microbiota composition (25, 38–41), and prior humanization studies have used donor-matched diets with the intent of improving engraftment (42) and recapitulation of the phenotype associated with the human donor (26). We observed that the fixed-formula mouse chow did not impair the engraftment of the Malian fecal samples and, interestingly, resulted in a relative abundance of certain taxa, particularly that of *Bacteroides* (Fig. 3A), closer to the input than that which was produced by the donor-matched diet. The chow-fed mice may have had a lower percentage of *Bacteroides* due to the presence of animal products in the chow, in contrast to the completely plant-based Malian diet, similar to what was observed by Turnbaugh *et al*., when mice were fed a Western diet (25).

In addition to *Bacteroides*, another major difference in the microbiome composition between the different mouse groups was the prevalence of *F. prausnitzii* (Fig. 2H), a major producer of butyrate and a keystone bacteria in humans (43–45). Previous studies examining the engraftment of human microbiota in mice observed a substantial decrease in *F. prausnitzii* abundance (18, 21, 25) or were unable to detect *F. prausnitzii* in the recipient mice (32). Wrzosek *et al*. were only able to detect *F. prausnitzii* in mice that had received repeated rounds of fecal microbiota transfer and not before the day 28 time point (18), and Wos-Oxley *et al*. noted a decrease of *F. prausnitzii* from 16.85% in humans to 3.03% in the gavaged germ-free mice and were not able to detect *F. prausnitzii* in antibiotic-treated mice gavaged with the same donor (21). Finally, Turnbaugh *et al*. observed a decrease in *F. prausnitzii* abundance from the human donor (19.15%) compared with that from the gavaged germ-free mice (1.34%). However, they found that when the human sample was frozen before gavage, the abundance of *F. prausnitzii* in the mice (6.22%) was higher compared with that of the mice that received the same sample before storage at −80°C (25), possibly due to the loss of viability of competing bacterial species.

In our study, *F. prausnitzii* was found at an abundance of 12.79% in the human input and was detected in all mouse groups at all time points, but the abundance and dynamics of the engraftment were dependent on the gavage group (Table S3). The Malian fecal samples were all stored at −80°C before use, which may partially account for the ubiquity and enriched engraftment of *F. prausnitzii* in our study. The relative

abundance of *F. prausnitzii* decreased quickly in the groups that only received gavages within the first week (1DMD and 4DMD) compared with the groups that received weekly gavages (4WMD and 4W7C). Within the groups receiving weekly gavages, the group of mice fed the fixed-formula mouse chow maintained levels of *F. prausnitzii* similar to that of the donor throughout the experiment, whereas the relative abundance decreased after the second week in the group fed the donor-matched diet. Stable engraftment of bacterial communities often requires cooperation within metabolic networks, including cross-feeding, wherein the metabolites of one species are utilized as substrates by others (46). *Akkermansia muciniphila* specializes in the degradation and fermentation of host mucus, which results in the liberation of mucus glycan sugars that have been shown to stimulate the coexistence of butyrogenic bacteria, such as *F. prausnitzii*, in co-culturing experiments (47). Cross-feeding by *A. muciniphila* may explain the stable engraftment of *F. prausnitzii* in the 4W7C group since *A. muciniphila* is only present in this group (Table S3), possibly due to the presence of animal products in the chow, which have been shown to be associated with elevated levels of *Verrucomicrobia* (24). *B. thetaiotaomicron* (48) and *Bifidobacterium adolescentis* (49) have also been shown to cross-feed *F. prausnitzii* in co-culturing experiments, but neither of these bacteria are likely to explain the sustained abundance of *F. prausnitzii* in the group fed the fixed-formula chow, as *B. thetaiotaomicron* was found in all groups and *B. adolescentis* was only found in the human input.

In previous engraftment studies (18, 21, 25), the fecal samples were prepared in anaerobic conditions and/or in reduced or nitrogen-sparged BHI or PBS, but this did not result in improved engraftment of *F. prausnitzii* relative to our study, despite our aerobic preparation of the Malian gavage sample. *F. prausnitzii* is a strict anaerobic bacteria that is very sensitive to oxygen and loses viability within 2 min of exposure to ambient air (43); however, *F. prausnitzii* can grow in the presence of oxygen if the growth media contain flavins and cysteine or glutathione (50), which are readily found in the intestinal environment (50). Thus, the presence of these molecules may explain why we observed enhanced engraftment *F. prausnitzii* compared with other studies that more extensively processed their stool samples before gavage. Furthermore, the repeated gavages at weekly intervals may have aided in the creation of a more favorable intestinal environment, including the adequate presence of flavins, cysteine, and glutathione, allowing for the improved colonization of *F. prausnitzii* observed in the two groups that received weekly gavages compared with the two groups that only received gavages during the first week.

To date, insufficient consideration has been given to the characterization of non-Western microbiota (35, 51), and as such, much less is known about non-Western gut microbiomes and their relationship with health and disease (22, 35–37). Nevertheless, there is increasing recognition of the crucial role that gut microbiota play in the etiology of numerous diseases and the immunogenic response to vaccination (13–16). Accordingly, accounting for the microbiome composition when studying disease pathology and developing clinical interventions will allow for the creation of more efficacious interventions and treatments for microbiota-associated diseases and for a broader understanding of the interaction of gut microbiota and well-being. Here, we compared the impact of the overall number and frequency of gavages on the humanization of germ-free mice using a rural African microbiome and whether using a donor-matched diet was necessary to establish adequate colonization. Overall, the gavage strategies used with Western microbiomes were broadly applicable to our Malian samples. We observed that one gavage was largely insufficient to provide stable engraftment of the microbiome and that the humanization resulting from multiple gavages was generally superior. Multiple gavages did not appear to increase the similarity of the microbiome to the input over time but rather to minimize the increase in distance from the input that occurred before the microbiome stabilized (Fig. S5; Tables S4 and S6). A donor-matched diet did not improve microbiome engraftment compared with a fixed-formula, polysaccharide-rich, low-fat mouse chow. Interestingly, the presence of a small amount of animal

product (90 g/kg fish meal) in the mouse chow may have improved the similarity of the engraftment of specific taxa to that of the Malian input microbiome. An animal product-containing Western diet has previously been associated with decreased *Bacteroides* within mice (25) and increased *Verrucomicrobia* (24). Thus, the fish meal in Inotiv/Envigo 7013 may have been responsible for the relatively reduced expansion of *Bacteroides* and the durability of *F. prausnitzii* colonization (through the presence of *A. muciniphila*) in the group that received the mouse chow.

A fixed-formula mouse chow was only used with the weekly gavage regimen; therefore, the exact impact that this diet would have had on the other two gavage regimens is unknown. The principal difference associated with using a fixed-formula mouse chow versus the donor-matched diet appears to be a difference in the relative proportion of *Bacteroides* and *Faecalibacterium*, which may be the result of the presence of dietary animal products. Differences in the relative abundances of *Bacteroides* species and the durability of the engraftment of *Faecalibacterium* were also responsible for most of the differences between the mice receiving multiple daily or weekly gavages. As such, it is possible that the use of the fixed-formula diet may have resulted in a similar colonization between mice receiving four daily or four weekly gavages. However, using a single gavage was associated with reduced stability and increased drift of the microbiome compared with using multiple gavages and is, therefore, unlikely to have been improved through the utilization of the fixed-formula mouse chow.

Further work toward understanding how certain factors (e.g., the presence or absence of supportive or competitive bacteria or dietary metabolites) modulate the microbiome composition following fecal transplant may allow for improved gut microbiota humanization and the creation of more effective murine models of human microbiota-dependent diseases. Moreover, additional attention needs to be given to non-American and non-European microbiomes to better understand the interaction between the gut microbiome and human health and to potentially improve the efficacy of treatments for gut microbiota-associated diseases in non-Western countries.

## MATERIALS AND METHODS

### Collection, preparation, and gavage of human stool samples

Stool samples were collected during a prospective cohort study conducted in Kalifabougou, Mali. The Ethics Committee of the Faculty of Medicine, Pharmacy and Dentistry at the University of Sciences, Techniques and Technology of Bamako, and the Institutional Review Board of the National Institute of Allergy and Infectious Disease, National Institutes of Health, approved this study (ClinicalTrials.gov identifier: NCT01322581). Written, informed consent was obtained from adult participants and from the parents and guardians of participating children. A detailed description of the cohort has been previously published (52). Aliquots of stool were cryopreserved at −80°C in Mali and shipped to the United States on dry ice for analysis. Five stool samples were randomly selected for this study to better capture the potential diversity of the stool microbiota. While on dry ice, a portion of each fecal sample was scraped off and diluted in sterile saline at 1:10 (wt/vol). The fecal suspension was vortexed, and the larger particles were allowed to settle on ice before combining an equal volume from each of the five separate fecal suspensions to create a single input. Five mice per group were gavaged with 200 µL of the fecal mixture at the indicated time points (Fig. 1A). The remaining gavage material was flash frozen in liquid nitrogen and stored at −80°C.

### Mouse husbandry

Germ-free, female C57BL/6N mice (5–7 weeks old) were purchased from Charles River Laboratories. Mice were housed in autoclaved Tecniplast IsoCage P cages (Tecniplast Group) with ALPHA-dri bedding (Shepherd Specialty Papers Inc.) and Bed-r'Nest nesting material (The Andersons Plant Nutrient Group) under a strict 12-hr light cycle. Cages

were changed once per week. The mice were provided *ad libitum* with either autoclaved 7013 (NIH-31 Modified Open Formula Mouse/Rat Sterilizable Diet) purchased from Inotiv/Envigo or a custom Malian diet (described below; Table S1A) and autoclaved reverse osmosis water. The cages were only opened within an Exspor-sterilized biosafety cabinet (40 min contact time) after being submerged in a tank of Exspor (5 min contact time). All items used with the mice (e.g., gavage needles) were autoclaved, and their packaging was wiped down with Exspor before being transferred into the biosafety cabinet. Fecal samples were collected upon arrival and from the sentinels at the end of the experiment, and sterility was verified by IDEXX BioAnalytics through generic bacteria 16S rRNA gene PCR and fungal and aerobic and anaerobic bacteria culture (case numbers 28000–2021 and 30724–2021). Fecal samples were collected from humanized mice at the indicated time points (Fig. 1A) and were flash frozen in liquid nitrogen and stored at −80°C. All animal experiments were carried out at Indiana University adhering to the local and national regulation of laboratory animal welfare, and all procedures were reviewed and approved by the Indiana University Institutional Animal Care and Use Committees (protocol numbers 19024 and 22010).

## Preparation of a representative Malian diet

Millet (Bob's Red Mill Millet Flour) and sorghum (Bob's Red Mill Sweet White Sorghum Flour) were purchased from Amazon.com Inc., red kidney beans (365 Organic Red Kidney Beans) and great northern beans (365 Organic Great Northern Beans) were purchased from Whole Foods Supermarkets, and kefir (Lifeway Organic Kefir Plain Unsweetened Whole Milk), sweet potato (48,16), kale (94,627), chard (93,264), collard greens (4,614), turnip greens (4,619), parsley (94,901), tomatoes (4,664), cabbage (4,069), red bell pepper (4,688), jalapeno peppers (4,709), serrano peppers (4,709), zucchini (4,067), eggplant (4,081), cucumber (4,062), mangoes (4,959) and papaya (3,111) were purchased fresh from The Kroger Company. Food was prepared in one 5-kg batch. Millet, sorghum, and kefir were used to make a thick porridge. The beans were soaked overnight and simmered until soft. The sweet potatoes were peeled, cut into small pieces, and simmered until soft. The kale, chard, collard greens, turnip greens, parsley, tomatoes, cabbage, red bell pepper, jalapeno peppers, serrano peppers, zucchini, and eggplant were blitzed in a food processor and cooked to make a thick relish. Mangoes, papayas, and cucumber were left uncooked. All non-porridge food items were blitzed in a food processor and then mixed thoroughly with the porridge. Food was portioned out into meal prep containers, frozen, then vacuum sealed, and stored at −20°C. Diet composition can be found in Table S1A. Bacterial composition of the Malian diet (Table S1B) was analyzed using the same method that was used for the fecal samples (described below). The bags containing the diet were warmed to room temperature and wiped down with Exspor before being brought into the biosafety cabinet. Fifty grams of Malian diet (per cage of five mice) was extruded into two petri dishes per cage once a day.

## Gut microbiota analysis

DNA was extracted from feces and gavage aliquots using the QIAamp PowerFecal DNA Kit (QIAGEN, Germantown, MD) according to the manufacturer's instructions. Sterile PBS and ZymoBIOMICS Microbial Community Standards (Zymo Research) were processed following the same protocol. DNA samples and ZymoBIOMICS Microbial Community DNA Standards (Zymo Research) were shipped overnight on ice packs to the Genome Technology Access Center (GTAC; Washington University, St. Louis, MO) for 16S rRNA gene sequencing using Multiple 16S Variable Region Species-Level Identification (MVRSION), an approach that sequences all nine hypervariable regions of the 16S rRNA gene with 14 primer pairs (53). The feature table was constructed by GTAC (53) and imported into QIIME2 for core diversity analysis. Statistical analyses were performed using QIIME2 (54) and GraphPad Prism Version 9.4.1 using default parameters for the analyses indicated in the figure legends.

## ACKNOWLEDGMENTS

We thank the residents of Kalifabougou, Mali, for participating in this study.

This work was supported by a grant from the National Institute of Allergy and Infectious Disease of the National Institutes of Health (R01 AI148525 to N.W.S.). Support provided by the Herman B Wells Center (NWS) was in part from the Riley Children's Foundation. The project described was supported by the Indiana University Health, Indiana University School of Medicine Strategic Research Initiative (N.W.S.). The Mali study was funded by the Division of Intramural Research, National Institute of Allergy and Infectious Diseases, National Institutes of Health. The content is solely the responsibility of the authors and does not necessarily represent the official views of the National Institutes of Health.

## AUTHOR AFFILIATIONS

[1]Department of Pediatrics, Ryan White Center for Pediatric Infectious Diseases and Global Health, Herman B. Wells Center for Pediatric Research, Indiana University School of Medicine, Indianapolis, Indiana, USA

[2]Department of Bioinformatics and Biostatistics, University of Louisville, Louisville, Kentucky, USA

[3]Malaria Infection Biology and Immunity Section, Laboratory of Immunogenetics, National Institute of Allergy and Infectious Diseases, National Institutes of Health, Rockville, Maryland, USA

[4]Mali International Center of Excellence in Research, Malaria Research and Training Center, University of Sciences, Techniques and Technologies of Bamako, Bamako, Mali

## AUTHOR ORCIDs

Kristin M. Van Den Ham  http://orcid.org/0009-0009-6766-4251
Nathan W. Schmidt  http://orcid.org/0000-0001-7325-4536

## FUNDING

| Funder | Grant(s) | Author(s) |
| --- | --- | --- |
| HHS | National Institutes of Health (NIH) | R01 AI148525 | Nathan W. Schmidt |

## DATA AVAILABILITY

The data sets generated and analyzed during the current study are available in the NCBI Sequence Read Archive: PRJNA939242.

## ADDITIONAL FILES

The following material is available online.

### Supplemental Material

**Supplemental figures and tables (Spectrum01554-23-S0001.pdf).** Fig. S1 to S7 and Tables S1 and S4 to S6.
**Supplemental Table S2 (Spectrum01554-23-S0002.xls).** Genus-level taxonomy.
**Supplemental Table S3 (Spectrum01554-23-S0003.xls).** Species-level taxonomy.

### Open Peer Review

**PEER REVIEW HISTORY (review-history.pdf).** An accounting of the reviewer comments and feedback.

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
