## [Reviewer comments · Microbiology Spectrum]

Microbiology Spectrum

Creation of a non-Western humanized gnotobiotic mouse model through the transplantation of rural African fecal microbiota

Kristin Van Den Ham, Morgan Little, Olivia Bednarski, Elizabeth Fusco, Rabindra Mandal, Riten Mitra, Shanping Li, Safiatou Doumbo, Didier Doumtabe, Kassoum Kayentao, Aissata Ongoiba, Boubacar Traore, Peter Crompton, and Nathan Schmidt

Corresponding Author(s): Nathan Schmidt, Indiana University School of Medicine

Review Timeline:

Submission Date:	April 12, 2023
Editorial Decision:	August 22, 2023
Revision Received:	August 28, 2023
Accepted:	September 5, 2023

Editor: Kevin Theis

Reviewer(s): Disclosure of reviewer identity is with reference to reviewer comments included in decision letter(s). The following individuals involved in review of your submission have agreed to reveal their identity: Walaa K. Mousa (Reviewer #1)

Transaction Report:

DOI: <https://doi.org/10.1128/spectrum.01554-23>

August 22, 2023

Dr. Nathan W. Schmidt
Indiana University School of Medicine
Pediatrics
1044 West Walnut Street
Indianapolis, IN 46202

Re: Spectrum01554-23 (Creation of a non-Western humanized gnotobiotic mouse model through the transplantation of rural African fecal microbiota)

Dear Dr. Schmidt:

Thank you for submitting your manuscript to Microbiology Spectrum, as well as your patience in this process. I recognize it was a uniquely long process. The manuscript has been reviewed by two investigators in the discipline. Both were supportive of the study and the manuscript. As you will see below, the comments are minor. Regarding Reviewer 1's comments on sample homogenization, processing, and donor choice, you may choose to address one or all of them in a Response to Reviewers alone or through brief explanations in the Materials and Methods section. Reviewer 2's comments were each minor and technical.

Link Not Available

Sincerely,

Kevin R. Theis

Journals Department
Reviewer comments:

Reviewer #1 (Comments for the Author):

I found the research interesting and impactful.

Comments:

Why the authors decided to mix the 5 fecal samples and did not use them as independent microbiome communities? I think one combined sample is not representative of the variation in the microbiome of the given population

What are the preexisting conditions of the donors? why the authors chose random patients? preexisting conditions have huge impact of diversity of gut microbes.

How mixing samples would interfere with interpretation of the results?

authors performed the microbiome recovery under anaerobic conditions? why only saline was used? the authors could try to grow the collection in vitro in a supplemented Gut microbiome growth media for 24-48 hrs before transferring to the mice to enhance the growth before these microbes got lost in the GIT of the recipient animal and give them better chance of colonization.

Minor comments

Writing; overall decent but some terms are not common such as "faithful colonization"

Reviewer #2 (Comments for the Author):

The manuscript aims to evaluate the optimal approach for establishing a humanized gnotobiotic mouse model based on fecal samples from a non-Westernized, Malian population. The number and frequency of gavages, as well as the influence of mouse diet, were considered. The manuscript is informative, well written, and there has been clear attention to detail in the study design and its communication.

I have only minor comments and suggestions for the authors.

Abstract

Line 40 - This is a very general statement. For the abstract, consider being explicit regarding "multiple" and "faithful". In the Introduction, this wording is fine.

Introduction

N/A

Methods

Place the Materials and Methods section after the Discussion per ASM guidelines.

Line 116 - Is there a specific approval number/ID that can be cited here? Same with line 145.

Line 121 - Should this be "scraped off"?

Line 147 - Is there a reference available for these ingredients representing a Malian diet, even if general and/or secondary? This is relevant to line 194 as well.

Line 175 - On what instrument was the sequence data generated? What specific protocols were used to process the sequence data? Minimally a reference which enumerates the process used.

Results

Line 207, 209, 210 - insert "the" before "mouse microbiome" (see also label of Figure 2, legend of Figure S1, label of Figure S2, line 264)

Line 222, 270, 307, 457 - Use "mouse" or "murine" to be consistent?

Line 235 - First appearance of "PCoA", thus write out here

Figures 4-6 - Consider merging some or all elements into a multipanel figure

Figure 1B-I, Figure 3 - It is not clear what the dotted lines in the figure panels represent

Discussion

Line 407 - change "bacteria" to "bacterium"

Staff Comments:

Preparing Revision Guidelines

To submit your modified manuscript, log onto the eJP submission site at <https://spectrum.msubmit.net/cgi-bin/main.plex>. Go to

Author Tasks and click the appropriate manuscript title to begin the revision process. The information that you entered when you first submitted the paper will be displayed. Please update the information as necessary. Here are a few examples of required updates that authors must address:

Please return the manuscript within 60 days; if you cannot complete the modification within this time period, please contact me. If you do not wish to modify the manuscript and prefer to submit it to another journal, please notify me of your decision immediately so that the manuscript may be formally withdrawn from consideration by Microbiology Spectrum.

We would like to thank the reviewers for the time they have invested to carefully review our manuscript and the supportive comments they have provided. We would also like to thank the reviewers for the thoughtful and constructive comments they have provided to us to improve the manuscript. Please find below our responses to each of these comments and how the manuscript has been revised.

Reviewer comments:

Reviewer #1 (Comments for the Author):

I found the research interesting and impactful.

Comments:

Why the authors decided to mix the 5 fecal samples and did not use them as independent microbiome communities? I think one combined sample is not representative of the variation in the microbiome of the given population.

What are the preexisting conditions of the donors? why the authors chose random patients? preexisting conditions have huge impact of diversity of gut microbes.

How mixing samples would interfere with interpretation of the results?

The ideal conditions for engraftment likely vary between fecal samples, even from those within the same community and those from the same donor over time. While it would be ideal to have a custom designed gavage procedure for each individual fecal sample, it would be unfeasible to do this with the number of samples required to obtain statistically significant results in human microbiome studies. Thus, we endeavored to design a generally applicable gavage procedure for non-Western fecal samples, or at the very least, for rural African samples, as this is our study population of interest. The use of a mixture of five samples was done with the hope of capturing a wider breath of the possible taxa present in this community than would be found within a single sample, while also creating a more representative sample that was less likely to be as influenced by deviancies present within the individual samples. While it is possible that the use of five individual samples (each given to five mice in each of the four separate groups) may have shown that some of the individual samples engrafted slightly better with another gavage regimen, it is likely that the consensus regimen would have been the same. The choice of randomized patients was done as there is a lack of information regarding what constitutes a “normal” microbiome in rural African population, thus choosing samples to capture the known variability based on the limited information that we currently possess likely would have inadvertently biased our fecal sample.

The fecal samples that were chosen came from children with no known pre-existing conditions. The children that gave the fecal samples were enrolled into the larger study population in May 2011, therefore we have information obtained from their clinical visits for the three years prior to the start of our study arm of interest in 2014. The children previously went to the clinic for various injury- and illness-related issues (toe dislocation, heel wound, fever, chills, headache, diarrhea, vomiting, conjunctivitis, tonsil infection, ear inflammation, pneumopathy, etc.).

However, none of the children were currently experiencing a symptomatic illness when they gave the samples that were used in our study. Two of the five samples came from children that were positive for *Plasmodium falciparum* by PCR and asymptomatic. Asymptomatic carriage of *P. falciparum* is very common in this population of children, and our current data shows that asymptomatic infection with *P. falciparum* does not significantly change the gut microbiome in these children compared to the gut microbiome of uninfected children.

authors performed the microbiome recovery under anaerobic conditions? why only saline was used? the authors could try to grow the collection in vitro in a supplemented Gut microbiome growth media for 24-48 hrs before transferring to the mice to enhance the growth before these microbes got lost in the GIT of the recipient animal and give them better chance of colonization.

The fecal gavage mixtures were not created under anaerobic conditions. The samples were collected at a rural clinic without the capacity to manipulate the samples in anaerobic conditions, thus the samples were previously exposed to aerobic conditions before we received them. Furthermore, we observed excellent colonization of known strict anaerobes (such as *Faecalibacterium prausnitzii*), which suggests that the limited exposure of the samples to oxygen had minimal impact on the ability of these anaerobes to colonize the mice. The samples were made fresh at each time point immediately before being gavaged into the recipient mice. Since the microbes were spending a limited amount of time in the solution before gavage, they did not need additional media supplementation and we wanted to avoid modifying the original conditions of the fecal sample as much as possible. For that reason, we also administered the fecal solution directly, without an intermediate incubation step. Growing the samples *in vitro* would likely enhance the prevalence of some of the microbes, but many microbes are unable to be cultured outside of the host and this culturing step would likely decrease the abundance of these microbes, if not completely eliminate their viable population from our sample, drastically modifying the composition of the gavage solution administered to the mice. Additionally, our lab, and many other labs, have observed that microbes initially present at very low amounts within the original sample can greatly expand once within a mouse and microbes present at relatively elevated levels can be lost, thus the success of colonization is likely more dependent on the host conditions than on their original abundances within the gavage sample.

Minor comments

Writing; overall decent but some terms are not common such as "faithful colonization"

The two instances of "faithful colonization" was changed to more standard terms.

Reviewer #2 (Comments for the Author):

The manuscript aims to evaluate the optimal approach for establishing a humanized gnotobiotic mouse model based on fecal samples from a non-Westernized, Malian population. The number and frequency of gavages, as well as the influence of mouse diet, were considered. The manuscript is informative, well written, and there has been clear attention to detail in the study design and its communication.

I have only minor comments and suggestions for the authors.

Abstract

Line 40 - This is a very general statement. For the abstract, consider being explicit regarding "multiple" and "faithful". In the Introduction, this wording is fine.

“multiple” and “faithful” were changed to be more explicit.

Introduction

N/A

Methods

Place the Materials and Methods section after the Discussion per ASM guidelines.

Materials and Methods were moved to after the Discussion.

Line 116 - Is there a specific approval number/ID that can be cited here? Same with line 145.

All relevant protocol and ID numbers for the clinical study can be found in the trial listed at ClinicalTrials.gov. This position of this reference in the relevant section has been moved slightly to make this clearer. The protocol numbers for the mouse work have been added.

Line 121 - Should this be "scraped off"?

Scrapped was changed to be scraped.

Line 147 - Is there a reference available for these ingredients representing a Malian diet, even if general and/or secondary? This is relevant to line 194 as well.

The ingredients used were based on informal observations at the study site and the relative amounts were based on the average of nutritional studies conducted in Mali, as indicated in the legend of Table S1. If needed, the emails from our collaborators regarding the ingredient composition of the typical meals eaten at the study site can be included as text in the supplementary file.

Line 175 - On what instrument was the sequence data generated? What specific protocols were used to process the sequence data? Minimally a reference which enumerates the process used.

All information regarding the instrument used to generate the sequence data and the protocols used to generate the OTU table can be found in the citation listed for MVRSION. The citation was added again in the Material and Methods section after reference was made to the construction of the OTU table by GTAC to clarify this point. Further clarification about the analysis of the data on our end has been included in the relevant section of the Materials and Methods.

Results

Line 207, 209, 210 - insert "the" before "mouse microbiome" (see also label of Figure 2, legend of Figure S1, label of Figure S2, line 264)

“the” added where appropriate.

Line 222, 270, 307, 457 - Use "mouse" or "murine" to be consistent?

These words are synonyms in the instances in which they are used.

Line 235 - First appearance of "PCoA", thus write out here

The definition of PCoA (principal coordinate analysis) was added to the first appearance of PCoA.

Figures 4-6 - Consider merging some or all elements into a multipanel figure

This can be done if the journal would prefer this organization. Otherwise, we would prefer to keep the figures as they are.

Figure 1B-I, Figure 3 - It is not clear what the dotted lines in the figure panels represent

The dotted line represents the level present in the input. This is indicated within the figure for Figure 1B-1 and Figure 3.

Discussion

Line 407 - change "bacteria" to "bacterium"

The word “bacteria” is in reference to *F. prausnitzii* as a taxa, not an individual bacterium.

September 5, 2023

Dr. Nathan W. Schmidt
Indiana University School of Medicine
Pediatrics
1044 West Walnut Street
Indianapolis, IN 46202

Re: Spectrum01554-23R1 (Creation of a non-Western humanized gnotobiotic mouse model through the transplantation of rural African fecal microbiota)

Dear Dr. Nathan W. Schmidt:

Your manuscript has been accepted, and I am forwarding it to the ASM Journals Department for publication. You will be notified when your proofs are ready to be viewed.

Sincerely,

Kevin Theis
Editor, Microbiology Spectrum
